

# Does forest replacement increase water supply in watersheds? Analysis through hydrological simulation

Ronalton Evandro Machado[1], Milena Lopes[1], Lubienska Cristina Lucas J. Ribeiro[1]

[1]Unicamp (University of Campinas) - School of Technology. Street Paschoal Marmo 1888, CEP 13484-332, Jd. Nova Itália,
Limeira, SP, Brazil.

*Correspondence to*: Ronalton Evandro Machado (machado@ft.unicamp.br)

**Abstract.** Forests play an important role in watershed hydrology, regulating the transfer of water within the system. Their role in maintaining the hydrological regime of watersheds is still a controversial issue. Consequently, we use the Soil and Water Assessment Tool (SWAT) model to simulate scenarios of land use in a watershed. In one of these scenarios we identified, through GIS techniques, "Environmentally Sensitive Areas" (ESAs) which have been undergoing watershed degradation and we considered these areas as protected by forest cover. This scenario was then compared to the current usage scenario regarding watershed sediment yield and hydrological regime. The results showed a reduction in sediment yield of 54% among different scenarios, whereas watershed water yield was reduced by 19.3%.

## 1 introduction

Knowledge on how forests affect the various aspects of water is essential to assess the role of forest cover on watersheds' hydrological regime (LIMA, 2010). Forests are often regarded as effective to stabilize and maintain the river flow rates and this is one of the reasons why revegetation is repeatedly recommended to recover watersheds (BACELLAR, 2005). Some of the hydrological functions usually ascribed to forests, however, such as increase rivers water availability, are disputable and lack a technical and scientific basis. We observe, however, that this is still a worldwide controversy, especially regarding the establishment of water conservation and sustainable use of natural resources policies.

In this line of research we find a large collection of data in the scientific literature, resulting from the systematic monitoring of watersheds all over the world, which use three methodologies, of which "paired basins" stands out (Brown, 2005). Some experiences with paired basins showed the effect of forest cover on water yield, where natural vegetation has been removed and/or replaced by planted forests (BOSCH and HEWLETT, (1982); BRUIJNZEEL (1990, 2004); BUYTAERT et al., (2006)).

The paired-basin technique would be arguably the best methodology to evaluate the hydrological functions normally assigned to forests, applicable to basins with very similar characteristics. It is always preferable that paired watersheds should be as near as possible, so as to have similar physical aspects, climate, vegetation and use and occupation (BEST et al., 2003). Despite the advantages of using paired micro-basins to study the impact of vegetation changes on water yield, this kind of study takes



time, since a watershed's hydrological response to tree cutting or reforestation is a medium- to long-term process. It is also impossible to test other configurations of land management and use.

Another option to predict the impact of land-use changes on the quantity and quality of water in a watershed, e.g., vegetation replacement, is the use of hydrological models. According to Sun et al. (2006) mathematical models are probably the best tools

to analyze complex non-linear relationships between the water yield of forests and major environmental factors.

The large number of existing models applied to watersheds shows the advancement of this technology. There are many hydrological models that simulate the quality and quantity of water flow, each one with strengths and weaknesses which must be considered according to the user's needs and the characteristics of the study area. As an example, the Soil and Water Assessment Tool (SWAT) model allows great flexibility when configuring watersheds (PETERSON & HAMLETT, 1998).

The model was developed to predict the effect of different management scenarios in the quality and quantity of water, sediment yield and pollutant loads in agricultural watersheds (SRINIVASAN & ARNOLD, 1994). SWAT analyzes watersheds divided in sub-watersheds based on relief, soil and land use, preserving thus spatially distributed parameters of the entire watershed and homogeneous characteristics within the watershed.

The SWAT model is internationally recognized as a solid interdisciplinary watershed modeling tool, as demonstrated in annual

international conferences and papers submitted to scientific journals (KUWAJIMA et al., 2011). SWAT's many uses have shown promising results, e.g., hydrological assessments,  impacts of climate change, evaluation of best management practices, estimation of pollutant load, determining of the effects of land-use change, sediment yield, etc (SRINIVASAN & ARNOLD, 1994; ROSENTHAL et. al., 1995; CHO et al., 1995; MACHADO & VETTORAZZI, 2003; MACHADO et. al. 2003; KOCH et al., 2012; LESSA et al., 2014; ABBASPOUR et al., 2015; DECHMI & SKHIRI, 2013; LIU et al., 2015; ZHANG et al.,

2014; ROCHA et al., 2015; LIN et al., 2015).

Due to the uncertainty of forests' role in the quantity and quality of waters produced by rivers and the possibility of creating different scenarios that are difficult to test at watershed level, this paper's objective was first to identify "Environmentally Sensitive Areas" (ESAs) in the watershed under study and, subsequently, to simulate land use scenarios comparing them regarding sediment yield and hydrological regime.

## 2 Materials and methods

### 2.1 Area of study

Pinhal River's watershed is located between UTM coordinates 250,000 m and 275,000 m (S), 7,490,000 m and 7,520,000 m (N) (UTM Zone 23 S, central meridian 45° W). It consists of approximately 300 Km$^2$ (Figure 1). It has a tropical highland climate – Cwa, according to the Köeppen classification, with a hot and humid summer and cold and dry winter, and average

annual temperature of 25°C. Average annual precipitation is approximately 1,240 mm.

Sugarcane cultivation occupies most of the watershed area (42.3%), whereas citrus fruits cultivation occupies approximately 30% of the area. Much of the original forest vegetation has been destroyed in the process of land use and occupation, now



scattered along the watercourse banks (9%). The built-up area occupies 6.7%, located at the western side of Pinhal River's watershed. The predominant soils in Pinhal River's watershed are oxisols (72%) and cambisols (19%).

The Pinhal River is important for being the source of water for Limeira, state of São Paulo. The watershed has suffered in the past few decades from environmental degradation. The current situation may compromise this water source, if the process of degradation continues.

![Pinhal watershed map showing elevation, meteorological stations, precipitation stations, hydrologic stations, hydrography and location within Brazil, State of São Paulo and Piracicaba basin]

Figure 1: Locations of the Pinhal watershed and gauging stations.

## 2.2 The SWAT model and input data

SWAT is a distributed parameter model which simulates different physical processes in watersheds and which aims at analysing the impacts of changes in land use on surface and subsurface runoff, sediment yield and water quality in agricultural



watersheds that were not instrumented (SRINIVASAN & ARNOLD, 1994). The model operates on a daily basis and can simulate periods of 100 years or longer to determine the effects of management changes. It has been widely applied in hydrological modelling, water resources management and water pollution issues (DOUGLAS et al., 2010).

SWAT uses a command structure to propagate runoff, sediments and agrochemicals across the watershed. The model's components include hydrology, climate, sediments, soil temperature, crop growth, nutrient and pesticide loading, and agricultural management (ARNOLD et al., 1998). The hydrological component of SWAT includes subroutines of surface runoff, percolation, lateral subsurface flow, return flow of shallow aquifer and evapotranspiration.

SWAT uses a modified formulation of the Curve Number (CN) method (USDA-SCS, 1972) to calculate surface runoff. The Curve Number method relates runoff to soil type, land use and management practices (ARNOLD et al., 1995). Sediment yield is estimated using the Modified Universal Soil Loss Equation (MUSLE) (WILLIAMS & BERNDT, 1977).

The model requires as input data daily precipitation, maximum and minimum air temperatures, solar radiation, wind speed and relative humidity. Data were obtained from UNICAMP's School of Technology's weather station, located in Limeira, state of São Paulo, at UTM coordinates 251145 m (W) and 7503161 (S). Rainfall data were obtained from two other rainfall stations (Figure 1). Other data include cartographic layers: Digital Terrain Model (DTM), Land and Soil Use. Soil physical and hydraulic properties and crop phenological properties are stored in the model database. Table 1 summarizes the input data used in the study. Inputting data (layers and alphanumeric data) into SWAT is made via an appropriate interface. The interface (ARNOLD et al., 2012) was developed between SWAT and GIS ArcGis. The interface automatically divides the watershed in sub-watersheds from the DTM and then extracts input data from the layers and Geodatabase for each sub-watershed. The interface display the model outputs using ArcGis charts and tables. We divided the Pinhal River watershed in 25 sub-watersheds up to the runoff measuring station at UTM coordinates 266175 m (W) and 7496308 (S) (Figure 1).

Table 1. Data sources for the Pinhal watersheds and input data for SWAT model.

| Input data | Data description | scale | Data sources |
|---|---|---|---|
| Land use | Land-use classification - agricultural land, forest, pasture, urban and water | 25,000 | Coordenadoria de Planejamento Ambiental, Instituto Geológico, Secretaria do Meio Ambiente do Estado De São Paulo, 2013 |
| Soil | Soil types and physical properties | 100,000 | Instituto Agronômico de Campinas |
| Topography | Digital Elevation Map (DEM) | 10,000 | Instituto Geográfico e Cartográfico São Paulo |
| Hydrological and Meteorological | precipitation, minimum and maximum temperature, solar radiation, wind speed | Daily | ANA, FT |





## 2.3 Model evaluation

During the analysis period (2012 to 2014) calibration of model was not possible due to inconsistency in the observed data (the measuring station was constantly submerged during the operating period of a reservoir associated with a power station).

Despite the impossibility of calibrating the model for the Pinhal hydrographic basin, we used the hydrological regionalization

methodology to validate the behavior of the model (Vandewiele, 1995; Bardossy, 2007). Hydrological regionalization is a technique that allows transferring information between watersheds with similar characteristics in order to perform calculations, in places where there are no data on the hydrological variables of interest (Emam et al., 2016). This technique becomes a useful tool for water resource management, especially when applied to the most important instruments of the Brazilian water resource policy, the concession of water resources' use rights and charging for the use of water resources (Fukunaga et al., 2015).

According to Tucci (2005), hydrological information that can be regionalized can come in the form of variables, parameters or functions. Hydrological function represents the relationship between a hydrological variable and one or more explanatory or statistical variables, such as the flow-duration curve or the relationship between impermeable areas and housing density (Tucci, 2002). The flow-duration curve relates the flow or level of a river and the probability of flowing greater than or equal to the ordinate value, thus being a simple, but concise and widely used method to illustrate the pattern of flow variation over

time (Naghettini and Pinto, 2007).

For the construction of the flow-duration curve in this work, the series of simulated flows in the period from 2012 to 2014 was initially put in ascending order. This series was statistically divided into 10 equal intervals. For each interval, the number of flows was counted and the respective cumulative frequencies of the interval were calculated from highest to lowest. For comparison, in the same graph, we plotted and simulated the regionalized flows, according to the State Department of Water

and Electric Energy (DAEE – state entity responsible for granting concessions of water resources in the state of São Paulo), allowing the verification of sub or overestimation through the simulated curve. The Nash-Sutcliffe model's efficiency coefficient (Nash and Sutcliffe, 1970) was used to validate the simulation's results, in addition to the visual analysis of the regionalized simulated flow-duration curve (NSE). The NSE (Eq. 2) was used to compare the regionalized and simulated flows in intervals of 5 in 5% probability of occurrence of the flow-duration curve. NSE can range from -∞ to 1, where 1 is the optimal

value and values above 0.75 can be considered very good (Moriasi et al, 2007). NSE is calculated according to Eq. (1):

$$NSE = 1 - \frac{\sum_{i=1}^{n}(Q_{OBSi} - Q_{SIMi})^2}{\sum_{i=1}^{n}(Q_{OBSi} - \overline{Q}_{OBS})^2} \tag{1}$$

The PBIAS (Eq. 2) of the simulated discharge in relation to the regionalized were also used (Gupta et al., 1999).

$$PBIAS[\%] = \left( \frac{\sum_{i=1}^{n}(Q_{OBSi} - Q_{SIMi})}{\sum_{i=1}^{n}(Q_{OBSi})} \right) * 100 \tag{2}$$

Where, $Q_{OBSi}$ and $Q_{SIMi}$ correspond to the observed and simulated discharge, respectively, on day i (m³/s), and $\overline{Q}_{OBS}$

corresponds to the observed average discharge, in (m³/s), and n corresponds to the number of events.





## 2.4 Identification of Environmentally Sensitive Areas (ESAs)

The concept of "Environmentally Sensitive Areas" was created in industrialized countries approximately 30 years ago due to increased soil and water degradation and the degree of severity of degradation (RUBIO, 1995). Degradation is being caused

by uncontrolled forest destruction, water pollution, wind and water erosion, salinization and inappropriate management of cultivated and uncultivated soil (GOURLAY, 1998).

Environmentally Sensitive Areas (ESAs) are areas that contain natural or cultural features important for a functioning ecosystem. They may be negatively impacted by human activities and are vital to the long-term maintenance of biological diversity, soil, water, or other natural resource, in the local or regional context (NDUBISI et al., 1995). An environmentally

sensitive area may also be considered, in general, a specific and delimited entity with unbalanced environmental and socioeconomic factors, or not sustainable for that particular environment (GOURLAY, 1998). As an example, high sensitivity may be related to land use, which in certain cases causes soil degradation. Annual crops in areas where the relief is hilly, with declivity and shallow soils, have a high risk of degradation.

To identify ESAs in the Pinhal River watershed within the context of environmental degradation, we adapted the results from

Adami et al. (2012) and identified three types of ESAs: Critical, Fragile and Potential. Adami et al. (2012) made an environmental analysis of the Pinhal watershed via a Geographic Information System (GIS) using key indicators of relief, soil and land uses to determine the capacity of natural resources and environmental fragility. The empirical analysis of the environmental fragility methodology was used to identify areas that require more attention for improving environmental conditions. The procedures employed by the authors in their study are shown in Fig.2.





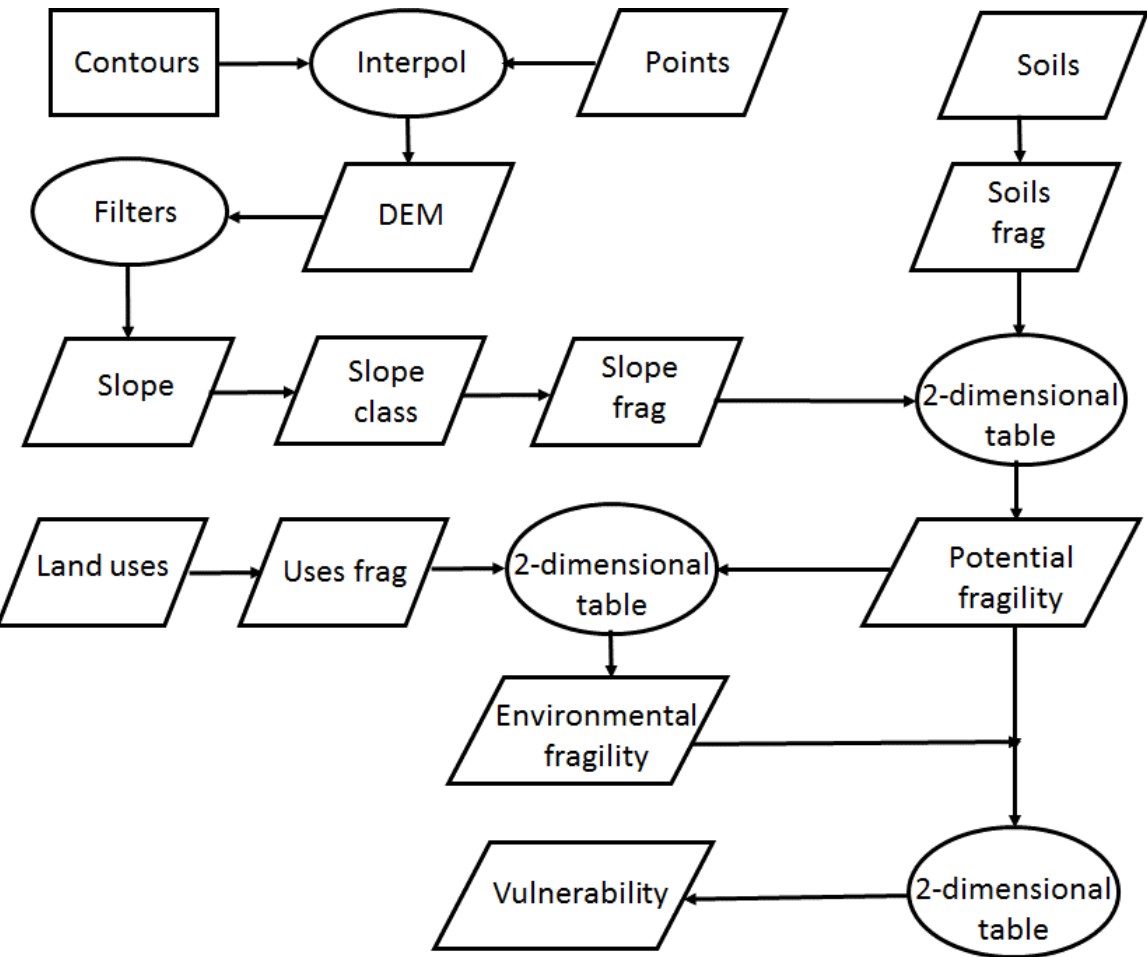

Figure 2. Flowchart of the procedures in the study by Adami et al. (2012).

## 2.5 Scenario simulation

We made two scenario simulations using the SWAT model interfaced with GIS ArcGis, aiming to verify the effect of land use change on sediment yield (sediment transported from sub-watersheds to the main channel over time, ton/ha) and the watershed hydrological regime (Discharge ($m^3$/s), surface runoff (mm), evapotranspiration (mm), soil water content (mm), water yield (mm)). Where the water yield (mm H2O) is the net amount of water that leaves the sub-basin and contributes to streamflow in the reach during the time step. (WYLD = SURQ + LATQ + GWQ – TLOSS – pond abstractions). SURQ is the surface runoff contribution to streamflow during time step (mm H2O). LATQ is the lateral flow contribution to streamflow during time step (mm H2O). GWQ is the groundwater contribution to streamflow (mm). Water from the shallow aquifer that returns to the reach during the time step. TLOSS is the average daily rate of water loss from reach by transmission through the streambed during time step ($m^3$/s) (ARNOLD et al., 2012).



One of the scenario simulations covered Critical and Fragile ESAs with overlapping forest cover on the land use map and we compared the results to the current scenario conditions (baseline). Thus, the land use pattern projected in this scenario is just hypothetical and often hard to implement in practice due to the already consolidated land use and occupation, but at the same time, it shows the watershed environmental fragility identified by Adami et al. (2012). Thus, these simulations illustrate the application and integration of hydrological and water quality models with GIS to evaluate watershed management scenarios, modifying only land use and occupation layer and management practices.

We used the deviation of the analyzed event (PBIAS) as statistical criterion to evaluate sediment yield and compare the hydrological behavior of the watershed in different scenarios, Eq. (3):

$$PBIAS[\%] = \left( \frac{\sum_{i=1}^{n}(Q_{CU} - Q_{ESA})}{\sum_{i=1}^{n}(Q_{CU})} \right) * 100 \tag{3}$$

Where, $Q_{CU}$ represents baseline scenario events (current use) in the period and and $Q_{ESA}$ the results of the alternative scenario (ESAs) in the period. Percent Bias calculation of the analyzed event (PBIAS) is important because it takes into account potential error among compared data. For this method, the higher the value of PBIAS (+ or -), the greater the difference in sediment yield and changes in hydrological regime among scenarios. Percent bias calculation of the analyzed event (PBIAS) is important because it takes into account potential errors in the compared data. For this method, the higher the value of PBIAS (+ or -), the greater the difference in sediment yield and changes in hydrological regime between scenarios.

## 3 Results and discussion

### 3.1 Model evaluation

Fig. 3 shows the discharge data obtained via regionalization and simulation (i.e., flow-duration curve). The flow-duration curves generated show that the simulation tends to underestimate the discharge almost uniformly, displaying greater differences in the probabilities of 20 to 100%, and overestimating only those with lower probabilities (10 to 20%). Despite underestimating flows most of the time, the simulated flow-duration curve displayed a pattern of variation similar to the pattern of variation of the regionalized flows. The NSE applied to compare the regionalized and simulated flows in intervals of 5 in 5% of the flow-duration curve was 0.93. According to Moriasi et al (2007), NSE values between 0.7 and 1 indicate a very good performance of the model. As for the PBIAS result for the flow values at intervals of 5% of probability of occurrence, the model underestimated the flows by 11%. PBIAS between 10% and 15% indicates a good accuracy of the model (Van Liew et al., 2007). Emam et al. (2016) used the SWAT model in the ungauged basin in Central Vietnam. The hydrological regionalization (i.e., ratio method) approach was used to predict the river discharge at the outlet of the basin. The model was calibrated with Nash-Sutcliff and $R^2$ coefficients greater than 0.7 in daily time scales by river discharge.



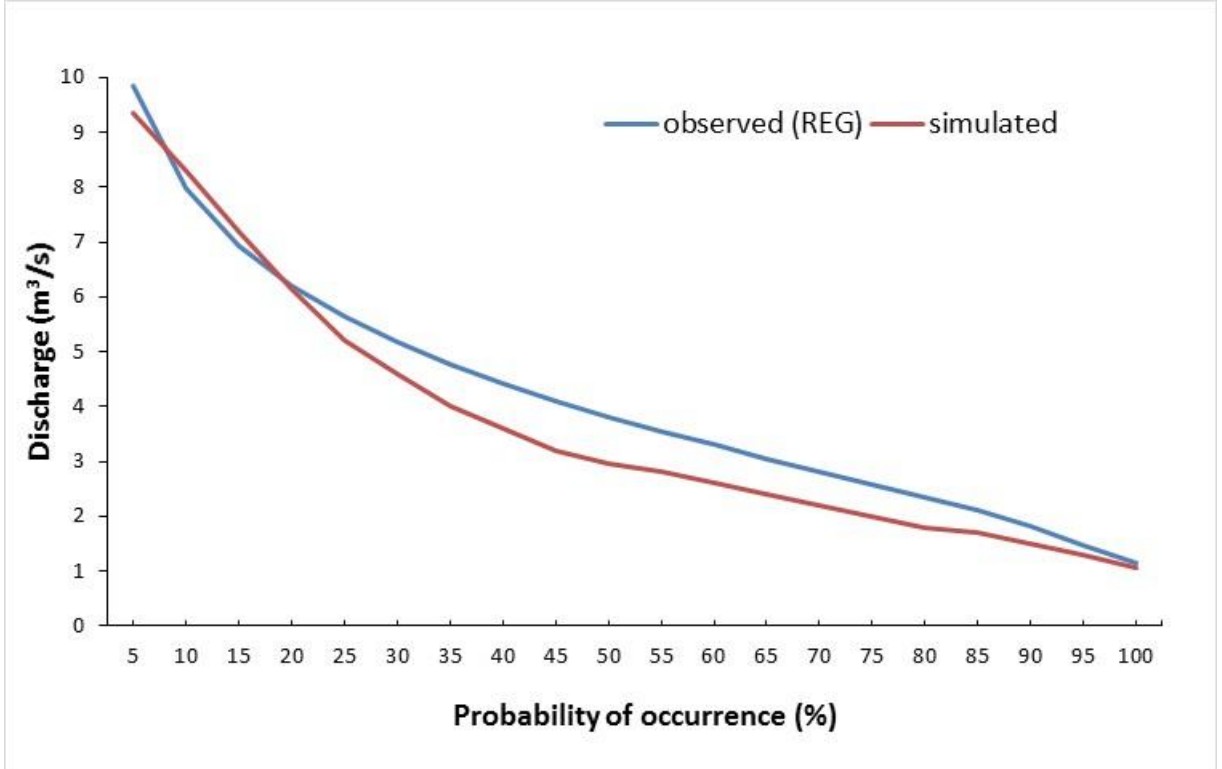

Figure 3 – Comparison of the observed (hydrological regionalization) flow-duration curve with the simulated one in the Pinhal watershed in the 2012-2014 period.

**3.2 Environmentally Sensitive Areas (ESAs)**

ESAs identified in the Pinhal River watershed are shown in Figure 4 and Table 2. 16% of the watershed area is degraded due to improper land use, which is a threat to the surrounding environment. These areas are severely eroded and have high rates of surface runoff and soil loss. In this case, there may be higher peak streamflow and sedimentation of water bodies (critical ESAs).

In 25% of the area we have identified regions where any change in the delicate balance between the environment and human

activities may result in environmental degradation of the ecosystem. A change in the soil management of annual and semiannual plants, e.g., sugarcane, in highly sensitive soils may cause an immediate increase in surface runoff and water erosion, pushing pesticides and fertilizers downstream (Fragile ESAs).

54% of the total watershed area is classified as Potential ESAs. Agricultural activities in these areas, although following Land Use Capability standards and requiring simple soil conservation practices to control erosion, require attention because of the

15 use of external agents such as pesticides in cultures of sugarcane and citrus fruits.





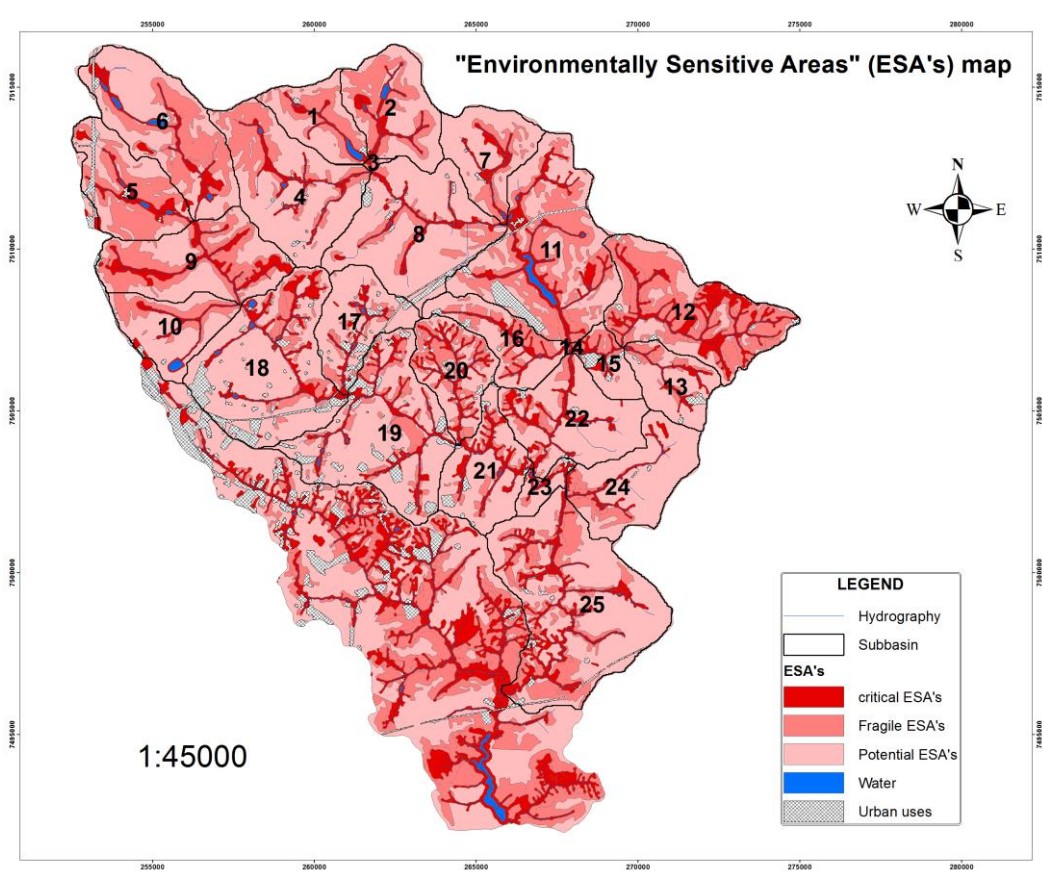

Figure 4 – ESAs Map in the Pinhal River watershed.

Table 2. Environmentally Sensitive Areas (ESAs) identified in the Pinhal River watershed.

| Class | Area (ha) | Area (%) |
|---|---|---|
| Critical ESAs | 4,801 | 16 |
| Fragile ESAs | 7,471 | 25 |
| Potential ESAs | 16,155 | 54 |
| Water | 149 | 1 |
| Urban or rural uses | 1,196 | 4 |
| Total | 29,772 | 100 |

**3.3 Land use change between scenarios**

5    Figure 5 presents the land use map for the two scenarios and Table 3, the total and relative areas of occupation of each land cover in the Pinhal River watershed for the current use scenario (baseline) and for the scenario of ESAs recomposed with native vegetation. From the current scenario to the ESAs' scenario there is a reduction of areas occupied with sugarcane, citrus





and pasture and, consequently, an increase of areas occupied with forest vegetation. Sugarcane occupied the largest area in the watershed and in the ESAs' scenario there was a reduction of 46.30% in this area. Orange occupies the second largest area in the current use scenario and in the new scenario it was reduced by 18.8%, whereas pasture was reduced by 44.43%. The area for other has been reduced by 42.61%. Some sub-watersheds increased forest cover compared to others: sub-watersheds

5    number 11, 12, 13, 14, 15 and 16.

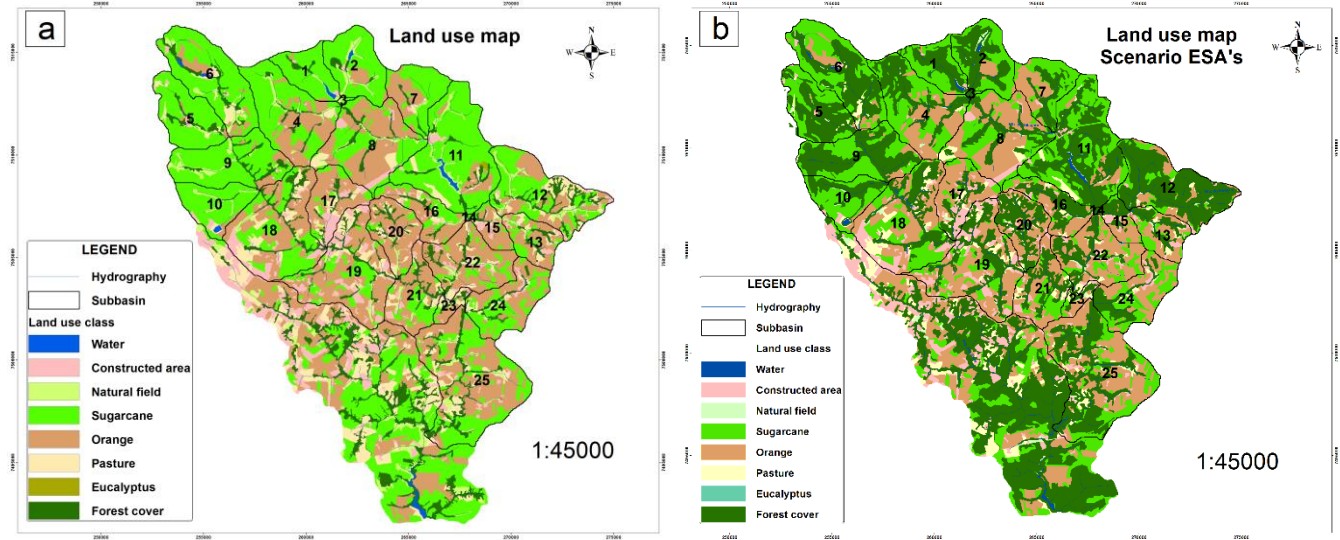

Figure 5 – Land use map for the current scenario (a) (Source: Secretaria do Meio Ambiente do Estado de São Paulo, 2013) and Critical and Fragile ESAs scenario (b), with native forest cover, overlapping current land use on the Pinhal River watershed (ESAs' scenario).

10   Table 3. Land use and occupation change between the two scenarios (current use and ESAs) in the Pinhal River watershed.

| Land-use type | Current use | | ESAs scenario | | Change | |
|---|---|---|---|---|---|---|
| | Area (ha) | Percentage (%) | Area (ha) | Percentage (%) | Area (ha) | Percentage (%) |
| Sugarcane | 12,566 | 42.2 | 6,748 | 22.7 | -5,818 | -46.30 |
| Orange | 8,866 | 29.8 | 7,199 | 24.2 | -1,667 | -18.80 |
| Pasture | 2,341 | 7.9 | 1,301 | 4.4 | -1,040 | -44.43 |
| Forest | 2,662 | 8.9 | 12,609 | 42.4 | 9,947 | 373.67 |
| Other uses | 3,337 | 11.2 | 1,915 | 6.4 | -1,422 | -42.61 |





We present in Figure 6 the variation of land use change in sub-watersheds scale between the two scenarios. The decrease in pasture and sugarcane areas, where soils are exposed to erosion during soil management, and the increase of native vegetation area, explain lower sediment yield and water yield. The decrease of pasture and increase of forest area in the Northwest region

5   (Sub-watershed 12) also contributed to lower sediment and water yield in this region.





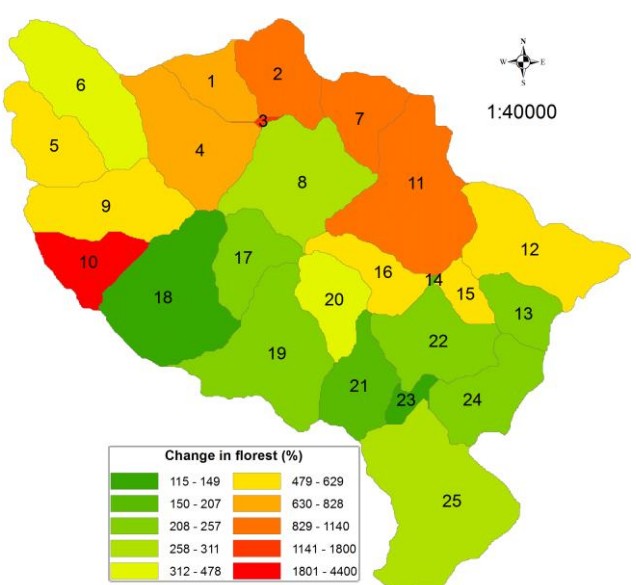

Figure 6. Spatial variations of land use types at sub-watershed scale between two scenarios.

## 3.4 Sediment Yield

The results of sediment yield presented in Figure 7 represent the erosion and sedimentation processes occurring throughout
Pinhal River watershed during the simulation period (2012 to 2014). With the scenario change, reduction in sediment yield
was -54% (PBIAS) compared to the current use scenario. This reduction occurred mostly in sub-watersheds located in lithosols
and cambisols (Figure 8). These are shallow, not deep soils. Cambisols in the watershed area occur in undulated relief. These
are poorly developed soils, with incipient B horizon. One of cambisols' main features is their shallowness and often high
content of gravel. High silt content and low depth are responsible for this low soil permeability (TERAMOTO, 1995). The
biggest issue, however, is soil erosion risk. Cambisols have restrictions of agricultural use, for their high erodibility, high risk
of degradation and poor trafficability. These soils occupy 19% of the watershed's total area. In the current use scenario, 22.4%
of this soil area is being occupied with native vegetation. In the ESAs' scenario this percentage increased to 68.3% (Table 4).
Lithosols occupy approximately 4% of the watershed's total area and are located in areas of greater declivity. They are in a
geomorphologically unstable zone in which erosion affects soil development, and they are constantly renewed through
superficial erosion (TERAMOTO, 1995). Extensive areas are occupied with sugarcane, pasture and orange (62.3%) cultivation
on these soils. In the current scenario, 24.3% of the lithosol is covered with native vegetation. In the ESAs' scenario this
percentage is 95.7% (Table 4). Increased native vegetation on both soils explains the 54% reduction (PBIAS) in sediment yield
in the watershed, when we compare the two scenarios. The spatial location of agricultural areas in relation to relief, soil and
climate is important to control erosion in watersheds.



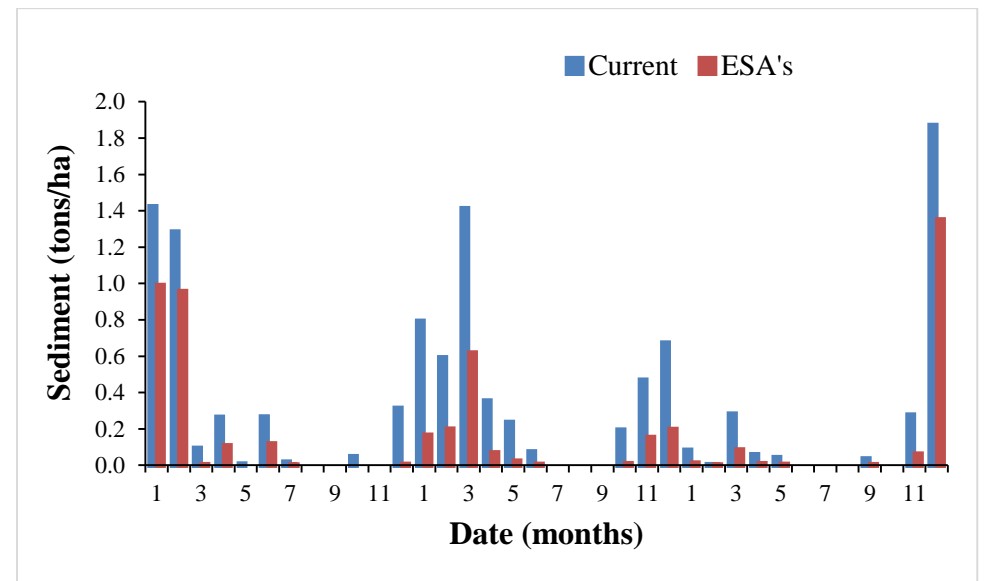

Figure 7 – Sediment yield comparison between the two scenarios on the Pinhal River watershed in the 2012-2014 period.

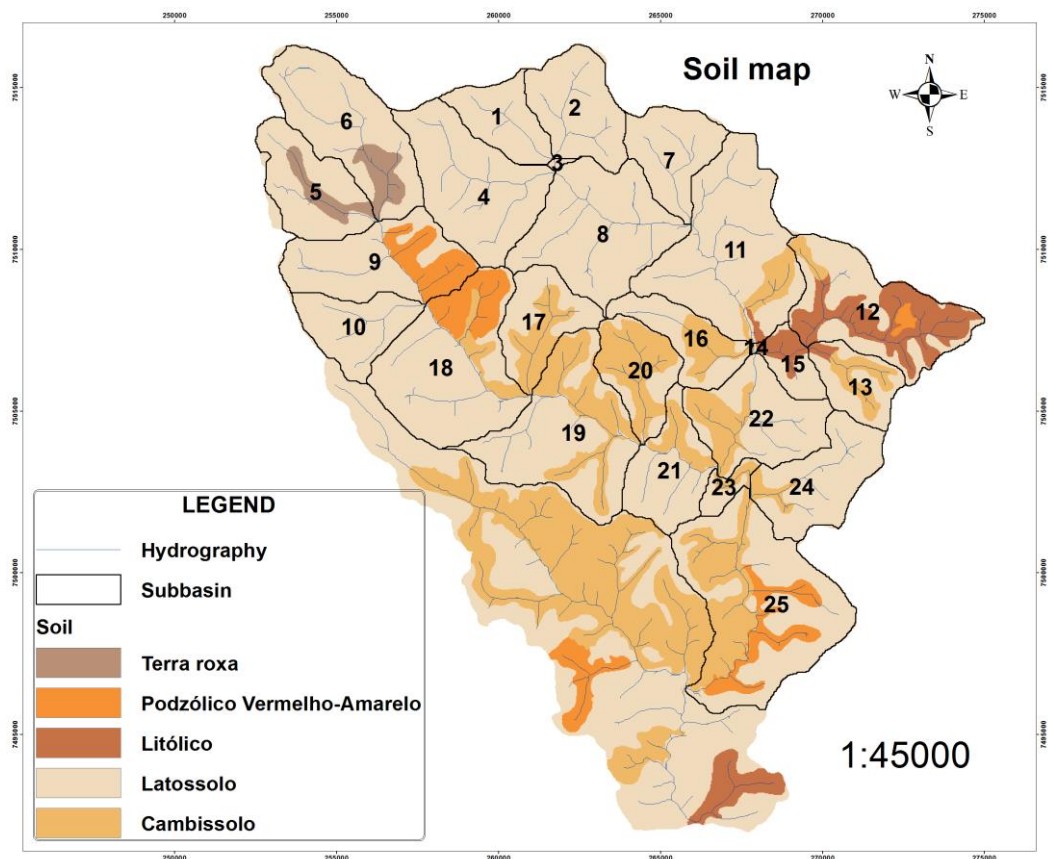

Figure 8 – Pinhal River watershed's soil map (Oliveira, 1999).





Table 4. Cross tab between land use changes in the scenarios for cambisols and lithosols in the Pinhal River watershed.

| Land use type | Cambisol | | | | Lithosol | | | |
|---|---|---|---|---|---|---|---|---|
| | Current use | | ESAs scenario | | Current use | | ESAs scenario | |
| | Area (ha) | Area (%) | Area (ha) | Area (%) | Area (ha) | Area (%) | Area (ha) | Area (%) |
| Forest | 1,278 | 22 | 3,894 | 68 | 275 | 24 | 1,089 | 96 |
| Pasture | 947 | 17 | 399 | 7 | 169 | 15 | 10 | 1 |
| Sugarcane | 997 | 17 | 142 | 2 | 350 | 31 | 8 | 1 |
| Other uses | 2,476 | 43 | 1,263 | 22 | 339 | 30 | 31 | 3 |

Spatially analysis of sediment yield in 25 sub-watersheds identified in the Pinhal River watershed's modeling (Figure 9) in the current use scenario showed a maximum sediment yield of 80.2 t/ha, with an average of 14.6 t/ha. Maximum sediment yield occurred in the upper Pinhal watershed, a more degraded area, whereas in the sub-watersheds in the lower Pinhal River watershed aggradation occurs, with lower sediment yield values. In the ESAs' scenario, replacement with native vegetation in Environmentally Sensitive Areas lead to an average sediment yield of 5.2 t/ha per year, with a maximum of 14.2 t/ha. Average soil loss in sub-watersheds was near tolerable soil loss rates, which according to Leinz & Leonardos (1977) is 7.9 ton/ha for podzol and 4.2 tons/ha for lithosol. According to Figure 7, the lowest rates of sediment yield occurred in sub-watersheds with greater forest cover. As the SWAT model simulates many processes in the watershed, some parameters may affect several processes (ARNOLD et al., 2012). With reduction of surface runoff in -45.8% (PBIAS) between scenarios (Table 5) due to greater soil protection, sediment yield has also been directly affected. Sediment yield difference between the two scenarios is presented in Figure 10. Analyzing Figure 10, this difference is greater in upstream sub-watersheds and in those with greater forest cover (sub-watersheds 11, 14, 15 and 16), according to Figure 5b.




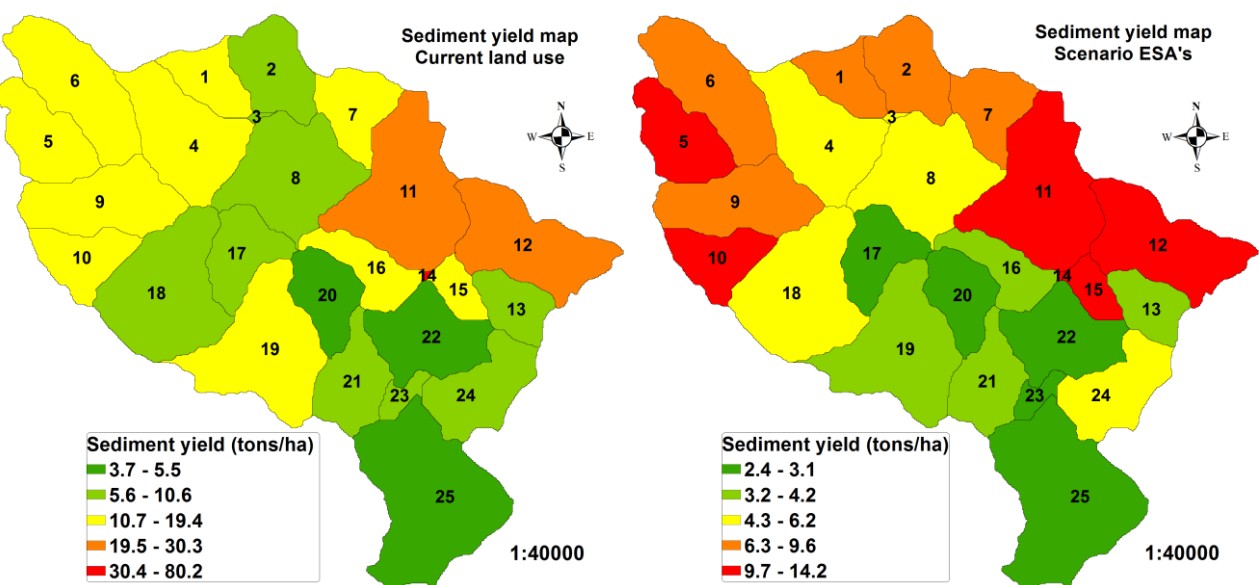

Figure 9 – Spatial distribution of average annual sediment yield at sub-watershed scale for the two scenarios.

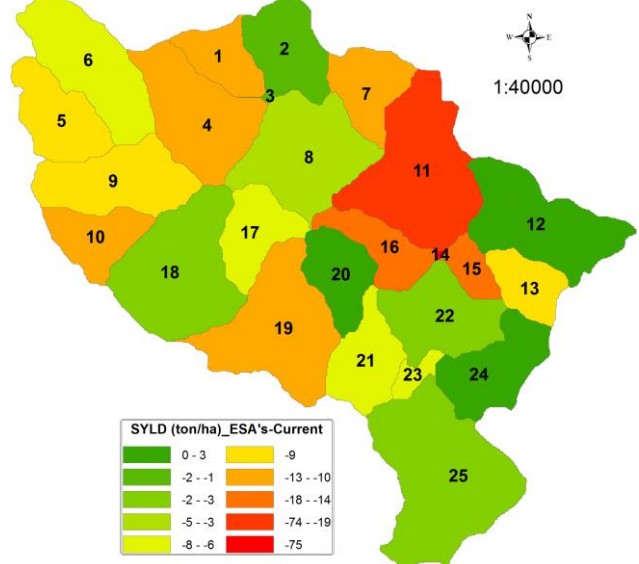

Figure 10 – Spatial variations of average annual sediment yield at watershed scale between the two scenarios.

## 3.5 Hydrological regime

It is widely reported that land use and land cover changes can affect the quantity and quality of water resources of a watershed. We analyzed the discharge (m3/s), surface runoff (mm), water yield (mm), evapotranspiration (mm) and soil water content (mm) (Figures 11-15) data to evaluate the impact of these changes on the watershed's hydrological regime. Monthly values for the 2012-2014 period were then compared between the two scenarios and the results (PBIAS) showed increased forest



cover in the watershed (+ 373.8%), decreased discharge, surface runoff (SR), soil water content (SW), water yield (WY) and increased evapotranspiration (ET) (Table 5). Studies conducted by Huang et al. (2003), Zhang et al. (2008), Li et al. (2009), Cui et al. (2012) showed that the increased forest cover in watersheds decreased water yield.

As both surface runoff and baseflow are the main components that contribute to water yield, we expected greater infiltration rate in the ESAs' scenario, for infiltration rate in forest areas is greater than in other land covers, e.g., sugarcane and pasture areas (Liu et al., 2012). Higher infiltration rate will increase baseflow, because in this scenario, areas previously occupied with other land uses were now occupied with native vegetation. On the other hand, forest evapotranspiration will consume more water (PBIAS of evapotranspiration equal to +3.5%, Figure 14), because it is known that the forest is the surface with highest rates of evapotranspiration, higher than all the other vegetation types and also higher than a liquid's surface (Birkinshaw et al., 2011). Roots, especially of larger trees, increase water absorption from the baseflow and, consequently, decrease water yield in the watershed, which may be seen in Figure 15, as the water content in the soil decreased in the studied period (-14.1%). Differently, with the scenario change, this type of land cover provides greater resistance to runoff and, consequently, this component had a lower contribution to water yield in the watershed (-45.8%).

Table 5. PBIAS of hydrological variables analyzed between the two scenarios (current use and ESAs) in the Pinhal River watershed, in the 2012-2014 period.

| Variable | Current use | ESAs scenario | PBIAS (%) |
|---|---|---|---|
| Discharge ($m^3/s$) | 119.1 | 105.3 | -11.6 |
| Surface runoff (mm) | 570.4 | 309.1 | -45.8 |
| Evapotranspiration (mm) | 1,993.2 | 2,062.3 | +3.5 |
| Soil water content (mm) | 8,279.8 | 7,113.5 | -14.1 |
| Water yield (mm) | 1,471.4 | 1,187.9 | -19.3 |

The influence of forest recovery in the hydrological regime can also be analyzed separately in two different periods. Comparing evapotranspiration demand independently in the wet period (October to March, Figure 14a) and dry period (April to September, Figure 14b), the difference between the two scenarios is even greater. In the wet period the difference is +1.3%, whereas in the dry period this difference is +8.2%. In the wet period, the available water in the soil (Figure 15a) compensates the increased evapotranspiration demand of vegetation, even with increased forest cover (ESAs' scenario), which contributes to lower water losses through evapotranspiration in the watershed (Figure 14a). In the dry period, when SW is lower (Figure 15b), large-sized forest vegetation access more easily underground water than small-sized vegetation, having, therefore, greater evapotranspiration demand and reducing water yield in the watershed. Based on results obtained from more than 90 experimental micro watersheds in different parts of the world, Bosch & Hewlett (1982) asserted that deforestation decreases





evapotranspiration, which results in more water available in the soil and in streamflow. On the other hand, reforestation decreases streamflow at watershed scale. It is worth mentioning, however, that these results vary from place to place and are often unpredictable (BROWN et al., 2005).

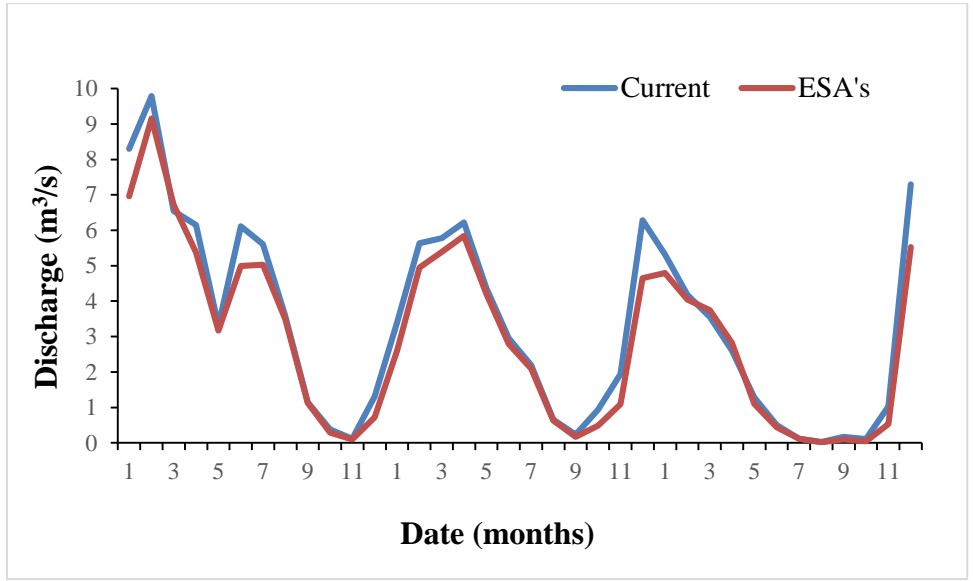

Figure 11 – Pinhal watershed streamflow comparison between the two scenarios.

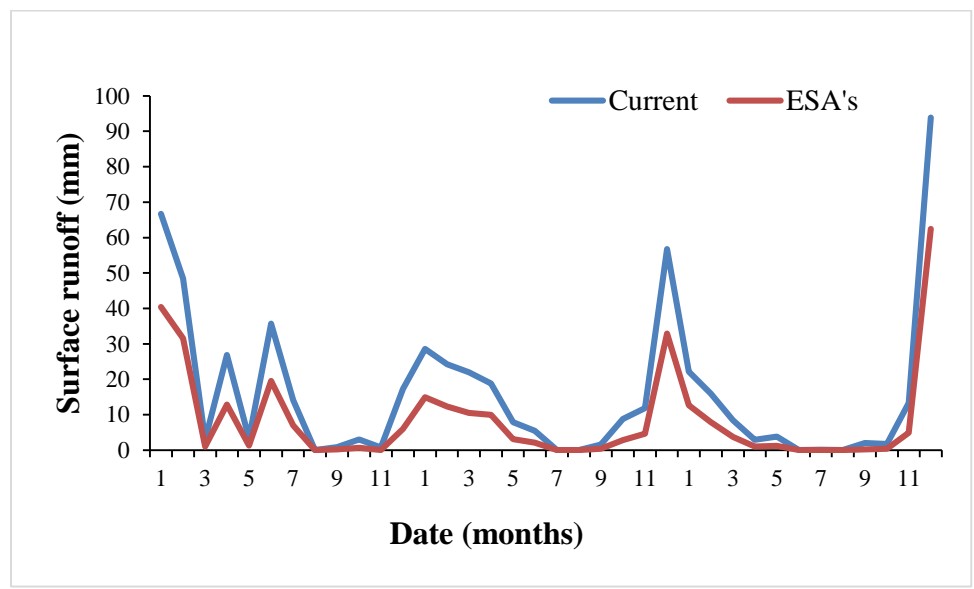

Figure 12 – Pinhal watershed surface runoff in the two scenarios.





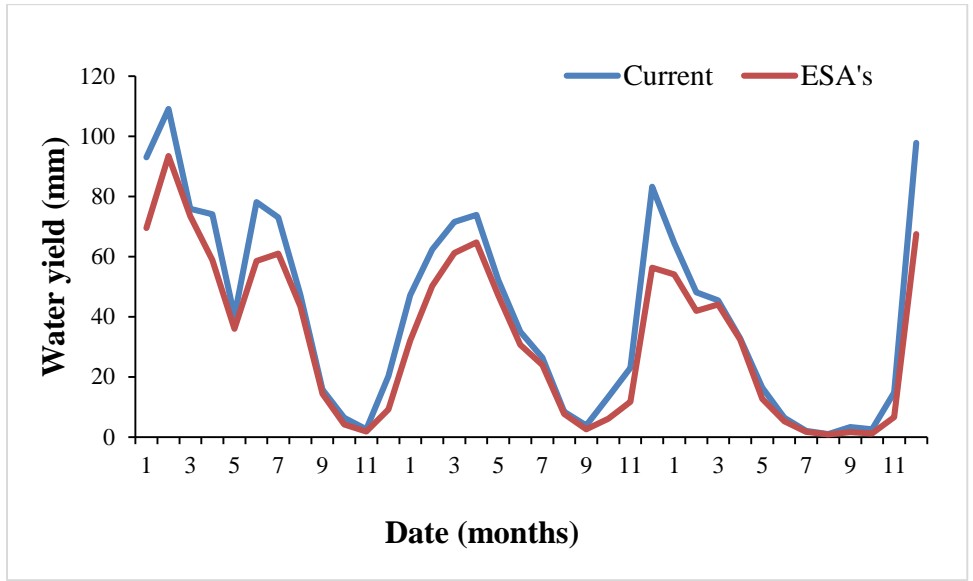

Figure 13 – Comparison of water produced in the Pinhal River watershed between the two scenarios.

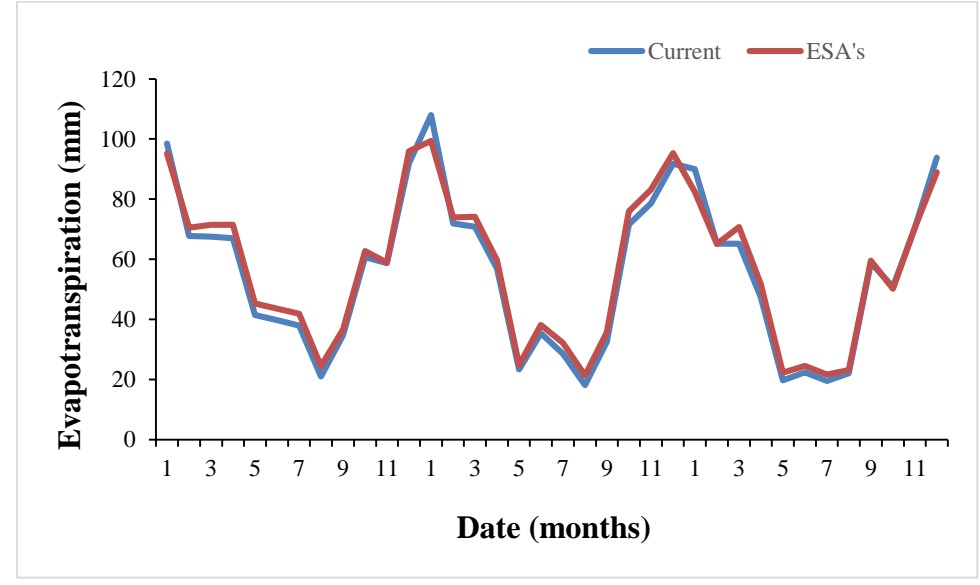

Figure 14 – Pinhal watershed evapotranspiration in two different scenarios.





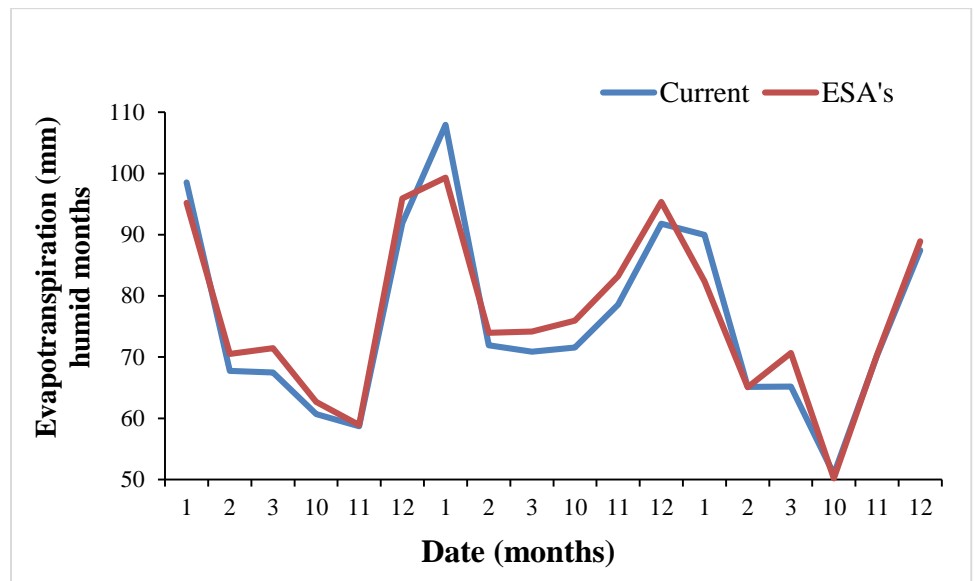

Figure 14a – Pinhal watershed wet season evapotranspiration in two different scenarios [PBIAS = +1.3%].

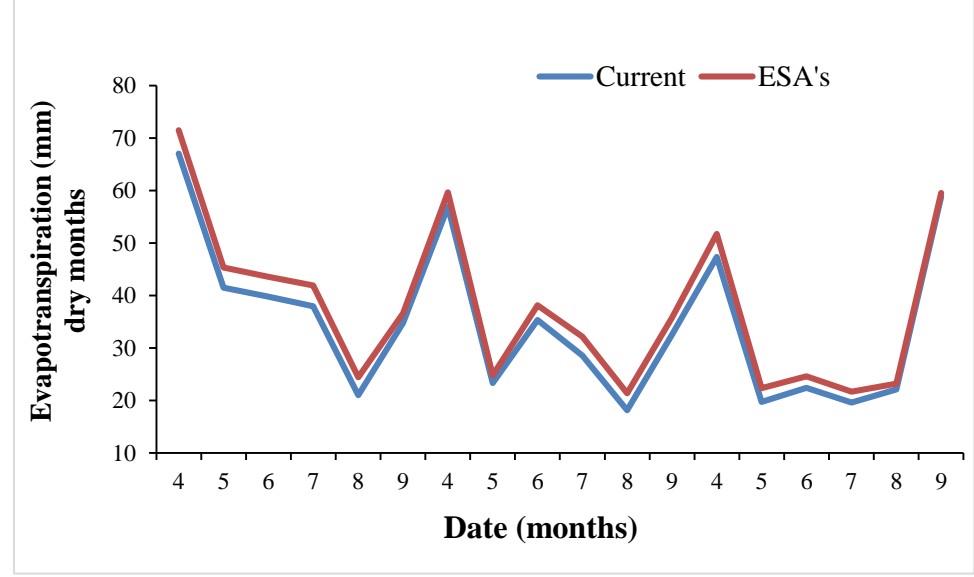

Figure 14b – Pinhal watershed dry season evapotranspiration in two different scenarios [PBIAS = +8.2%].





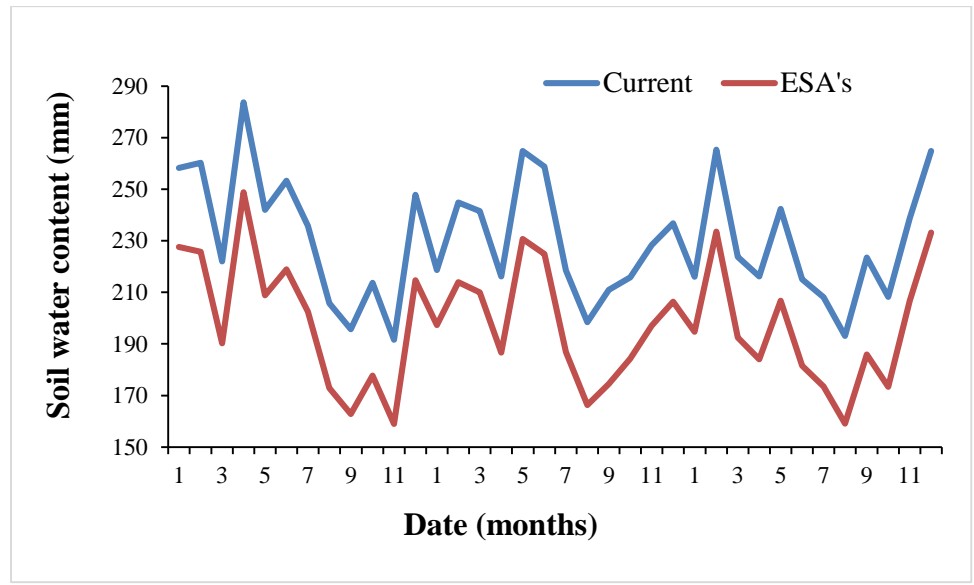

Figure 15 – Pinhal watershed soil water content in two different scenarios.

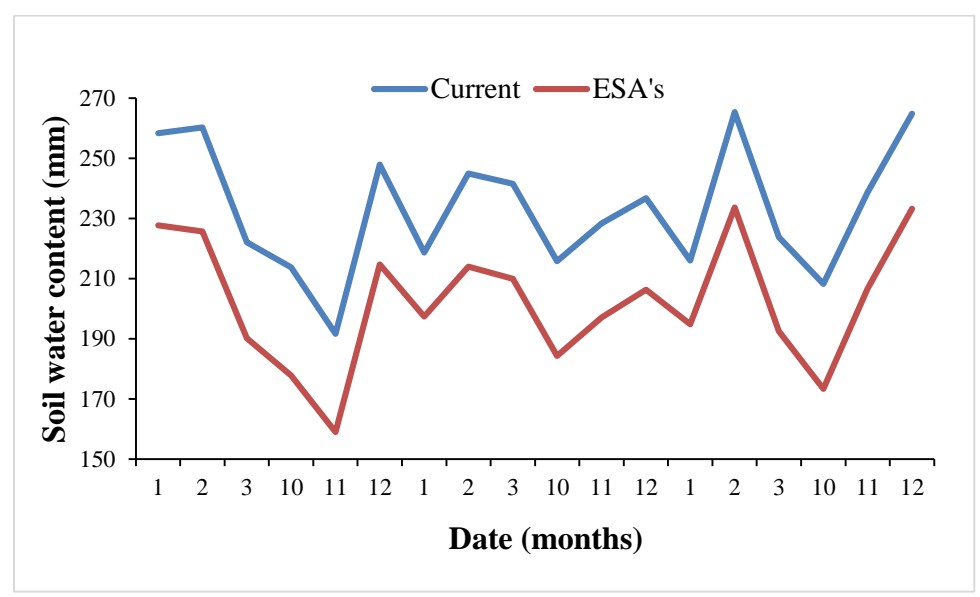

Figure 15a – Pinhal watershed soil water content in two different scenarios [PBIAS = -13.3%].





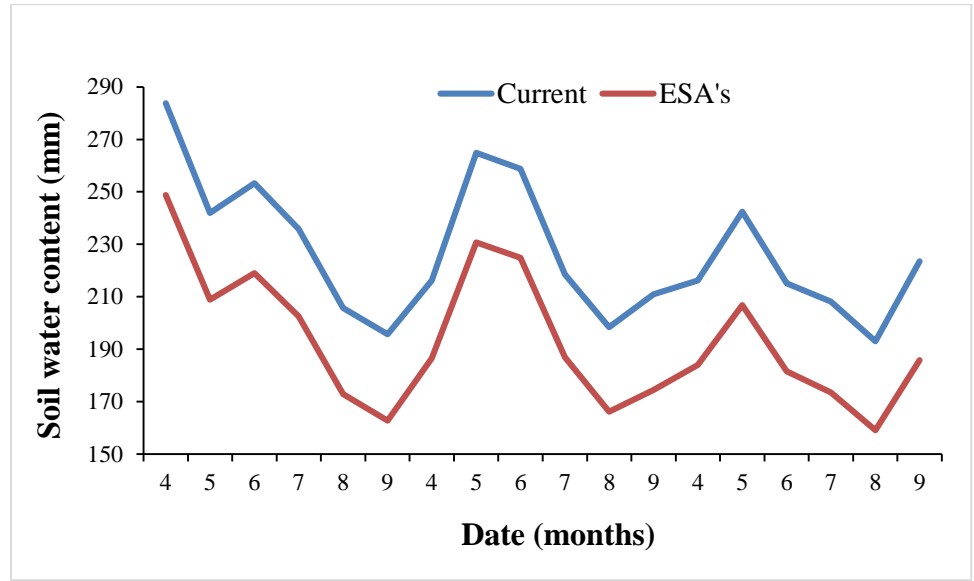

Figure 15b – Pinhal watershed soil water content during dry season in two different scenarios [PBIAS = -14.9%].

Figure 16 shows the spatial distribution of the hydrological regime variation (surface runoff, evapotranspiration, soil water content and water yields) at sub-watersheds scale between scenarios. The influence of land-use change on the hydrological

5   regime is more visible in some of the sub-watersheds than others at watershed scale. These variations were smaller in upstream sub-watersheds and as with sediment yield, major variations occurred in sub-watersheds with greater forest cover when we compare the current scenario with the ESAs' scenario. Watersheds' hydrological regime is the result of complex interactions between climate (wet versus dry years), plants' physiological properties (e.g., leaf area and successional stages) and soil type (ANDREASSIAN, 2004). According to Singh & Mishra (2012), these and other factors together make hydrological effects of

10   forests a markedly different scenario.





Figure 16. Spatial variations of the average annual hydrological regime at sub-watershed scale between the two scenarios. SURQ (surface runoff - mm), ET (evapotranspiration - mm), SW (soil water content - mm), WYLD (water yield - mm).

## 4 Conclusion

The role of forests in watersheds' hydrological cycle and water yield is controversial. Although reducing sediment yield as the results obtained from the simulation of different scenarios show (PBIAS = -54%), for it offers the soil greater protection, its influence on increasing and maintaining streamflow is questionable, because the results obtained from this study also showed that increased forest cover decreased water yield in the watershed in -19.3% (PBIAS) due mostly to its greater



evapotranspiration capacity (+3.5%), this demand being even greater during the dry season (+8.2%). Simulation results lead us to conclude that the impacts of land use change on hydrological processes are complex and their consequences are not equal in all situations and with the same intensity.

**Acknowledgments**

UNICAMP Espaço da Escrita project/General Coordination for the English translation of this article.

**Funding**

This work was funded by the São Paulo Research Foundation (FAPESP) [1grant #2013/02971-3].

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
