# Peer review of "Does forest replacement increase water supply in watersheds? Analysis through hydrological simulation"

_Hydrology and Earth System Sciences, 2017_

## Referee Comment (RC1) · Anonymous Referee #1 · 20 Jun 2017

The paper by Ronalton Evandro Machado et al. studies the effect of forest change on hydrologic regime in the Pinhal River watershed in Brasil. They used the SWAT model to simulate changes in water balance and sediment yield as results of changes in land use/cover scenarios in which one of them covered critical and fragile environmental sensitive areas (ESAs) with overlapping forest cover on the land use map. They concluded that the role of forest in hydrological process and water yield is controversial and that impacts of land use change on hydrological processes are complex and with various consequences.

I was very enthusiastic when I start to read the work but as I reach the end of the

manuscript I didn't find any novelty either to model development theory or hydrological modelling approach. In addition, i find that the paper doesn't add any additional knowledge to the role or the impact of forest change in the hydrological cycle.

Authors used regionalized stream flow to calibrate SWAT model but without any uncertainty assessment and compared the scenarios simulation the SWAT simulation. They didn't present the regionalization approach in the manuscript which is very important in that case. Further, authors tested a scenario where forest covers Critical and Fragile ESAs in the catchment but they admit that this is a theoretical scenario that cannot be realized in practice. So, what is the aim of testing such scenario? and how it can assist or inform the management of water in the catchment?

Some details L.27, P.2. add in Brazil L.30, P.3. 1,240 mm or 1240 mm?

P.4. Section: SWAT model and input data: Please update with recent SWAT references

P.5. Section Model evaluation. This section is not clear. The regionalization approach is not described. For instance, what kind of information are transferred (is it the FDC?) and what are the catchment attributes or similarity considered to perform the regionalization? This section also lacks discussion (at least few lines) regarding the uncertainty related to the regionalization technique.

P.5(2). Why not apply and calibrate SWAT in a physically similar catchment and then transfer the model parameters (calibrated) instead of transferring the FDC and then calibrate the model?

P.7. I don't see the need to report the flowchart by Adami et al. (2012) in this manuscript. I would prefer to see the flowchart of the methodology of the paper.

---

## Referee Comment (RC2) · Anonymous Referee #2 · 16 Jul 2017

General comments 1. The difference of water balance between 2 scenarios (current use and ESAs) is questionable. Based on results in Table 5 assuming similar annual rainfall, evapotranspiration increases 69.1 mm/y and water yield decreases 283.5 mm/y, its balance is -214.4 mm/y, but soil water content (or soil-water storage) is -1,166.3 mm/y. Basic water balance for sub-basin and the whole catchment should be re-checked.

Specific comments 1. P7L5-10: what is water balance equation in sub-basin? 2. P7L6: Partition of rainfall into Q, E, S are watershed water balance, not watershed hydrological regime. 3. P8L10-15: Some sentences are repeating written. 4. P8L22:

what is the meaning of "... flows in intervals of 5 in 5% of flow-duration curve...". 5. Figure 8: legend of lithosols and cambisols are not clear, check hydrography 's legend. 6. Table 4: adding one more row for total number. 7. P15L9: replace Figure 7 with Figure 5. 8. P17L1: increased forest cover in the watershed in Table 3 is 373.67%. 9. P17L5: How SWAT incorporate greater infiltration rate from more forest area? 10. P17L13: replace water yield (-45.8%) with (-19.3%). 11. Figure 14: its caption should be 14(a) whole , 14(b) wet 14(c) dry 12. Figure 15: similar to Figure 14.

---

## Author Comment (AC1) · 31 Jul 2017

**ANSWER**:

**Questions:**

**General comments:**

I'm sorry for frustrating your expectation as to the contribution of this paper. In Brazil we have heated discussions about the role of forests in the water regime in river basins. Although there are some publications in this line of research, some of the hydrological functions usually ascribed to forests, however, such as to increase rivers water availability are disputable and lack a technical and scientific basis. As the verification of these functions has a high cost and demand a long time, our objective was to bring to this discussion, the use of hydrological modeling tool. SWAT model outputs are less explored to reduce time, cost, and test various configurations in the watershed.

I rewrote the methodology of regionalization in the manuscript:

**2.3 Model evaluation**

[revised manuscript text omitted]

**P.3L.30** 1,240 mm or 1240 mm?

Is one thousand two hundred and forty. Minha dúvida é como representar 1,240 mm or 1240 mm.

Is one thousand two hundred and forty. My question is how to represent: 1,240 mm or 1240 mm?

**P.4.** Section: SWAT model and input data: Please update with recent SWAT references.

I updated SWAT references:

**2.2 The SWAT model and input data**

SWAT is a physically based semidistributed watershed-scale computationally efficient continuous-time hydrological model that operates on a daily/subdaily time step (Mohammed et

al., 2017). The SWAT model simulates different physical processes in watersheds and which aims at analysing the impacts of changes in land use on surface and subsurface runoff, sediment yield and water quality in agricultural watersheds that were not instrumented (Srinivasan & Arnold, 1994; Douglas et al., 2010; Ligaray et al., 2015). SWAT uses a command structure to propagate runoff, sediments and agrochemicals across the watershed. The model's components include hydrology, climate, sediments, soil temperature, crop growth, nutrient and pesticide loading, and agricultural management (Arnold et al., 1998). The hydrological component of SWAT includes subroutines of surface runoff, percolation, lateral subsurface flow, return flow of shallow aquifer and evapotranspiration (Grusson et al., 2017). SWAT uses a modified formulation of the Curve Number (CN) method (USDA-SCS, 1972) to calculate surface runoff. The Curve Number method relates runoff to soil type, land use and management practices (Arnold et al., 1995). Sediment yield is estimated using the Modified Universal Soil Loss Equation (MUSLE) (Williams & Berndt, 1977).

The model requires as input data daily precipitation, maximum and minimum air temperatures, solar radiation, wind speed and……………..

**P.5. and P.5(2).** Section Model evaluation.

To see "2.3 Model evaluation" above.

**P.7.** I don't see the need to report the flowchart by Adami et al. (2012) in this manuscript. I would prefer to see the flowchart of the methodology of the paper.

I reported the flowchart of Adami et al. (2012) by suggestion of the Handling Editor Xuesong Zhang. However, the paper is available in:

http://eduem.uem.br/ojs/index.php/ActaSciTechnol/article/viewFile/10005/pdf

---

## Author Comment (AC2) · 1 Aug 2017

**ANSWER**:

*Questions:*

*General comments:*

I did not analyze the water balance in the basin. For this, I would have to compute also, the PERC (water that percolates past the root zone) and GW_Q (Groundwater contribution to streamflow) – to see water balance equation in question 1 in specific comments below.

**Specific comments:**

**1. P7L5-10:** what is water balance equation in sub-basin?

P= ET + WYLD + SW + PERC - GW_Q (SWAT Documentation, version 2012)

Where:

P= Precipitation

ET= Evapotranspiration

WYLD= Water Yield

SW= Soil Water storage

PERC= water that percolates past the root zone

GW_Q= Groundwater contribution to streamflow

**2. P7L6:** Partition of rainfall into Q, E, S are watershed water balance, not watershed hydrological regime. 3.

As I did not analyze the hydrological balance in the basin but rather hydrological variables, I modified the expression "hydrological regime" for "hydrological processes".

**3. P8L10-15:** Some sentences are repeating written.

I rewritten the sentence:

Where, $Q_{CU}$ represents baseline scenario events (current use) in the period and and $Q_{ESA}$ the results of the alternative scenario (ESAs) in the period. Percent bias calculation of the analyzed event (PBIAS) is important because it takes into account potential errors in the compared data. For this method, the higher the value of PBIAS (+ or -), the greater the difference in sediment yield and changes in hydrological processes between scenarios.

**4. P8L22:** what is the meaning of "flows in intervals of 5 in 5% of flow-duration curve…".

According to Fig. 3, I compared the flows simulated by the model and those obtained by the hydrological regionalization at each interval of 5% of the probability of occurrence of the flow-duration curve.

I rewritten the sentence:

"The NSE applied to compare the regionalized and simulated flows at each intervals of 5% of the flow-duration curve was 0.93."

**5.** Figure 8: legend of lithosols and cambisols are not clear, check hydrography's legend.

I modified legend of Figure 8 accord to WRB (World Reference Base for Soils Resources).

[Figure]

Figure 8 – Pinhal River watershed's soil map (Source: Oliveira, 1999). Legend accord to WRB (World Reference Base for Soils Resources)

**6.** Table 4: adding one more row for total number.

I added one row for total number:

Table 4. Cross tab between land use changes in the scenarios for cambisols and lithosols in the Pinhal River watershed.

| Land use type | Cambisol | | | | Lithosol | | | |
|---|---|---|---|---|---|---|---|---|
| | Current use | | ESAs scenario | | Current use | | ESAs scenario | |
| | Area (ha) | Area (%) | Area (ha) | Area (%) | Area (ha) | Area (%) | Area (ha) | Area (%) |
| **Forest** | 1278 | 22.4 | 3894 | 68.3 | 275 | 24.3 | 1089 | 95.7 |
| **Pasture** | 947 | 16.6 | 399 | 7.0 | 169 | 14.9 | 10 | 0.9 |
| **Sugarcane** | 997 | 17.5 | 142 | 2.5 | 350 | 30.9 | 8 | 0.7 |
| **Other uses** | 2476 | 43.5 | 1263 | 22.2 | 339 | 29.9 | 31 | 2.7 |
| **Total** | 5698 | 100 | 5698 | 100 | 1138 | 100 | 1138 | 100 |

**7. P15L9:** replace Figure 7 with Figure 5.

I replaced Figure 7 by Figure 5.

**8. P17L1:** increased forest cover in the watershed in Table 3 is 373.67%.

I corrected the value for 373.67%.

**9. P17L5:** How SWAT incorporate greater infiltration rate from more forest area?

Land cover characteristics (SWAT Documentation, version 2012)

"The plant canopy can significantly affect infiltration, surface runoff and evapotranspiration. As rain falls, canopy interception reduces the erosive energy of droplets and traps a portion of the rainfall within the canopy. The influence the canopy exerts on these processes in a function of the density of plant cover and the morphology of the plant species."

**10. P17L13:** P17L13: replace water yield (-45.8%) with (-19.3%).

I replaced (-19.3%) by (-45.8%).

**11.** Figure 14: its caption should be 14(a) whole, 14(b) wet 14(c) dry 12.

I altered caption Figure 14

**12.** Figure 15: similar to Figure 14.

I altered Figure 15 like Figure 14.